# Fungus-Based Magnetic Nanobiocomposites for Environmental Remediation

**Thais de Oliveira Chaves [1], Raquel Dosciatti Bini [2], Verci Alves de Oliveira Junior [1], Andressa Domingos Polli [1], Adriana Garcia [1], Gustavo Sanguino Dias [2], Ivair Aparecido dos Santos [2], Paula Nunes de Oliveira [3], João Alencar Pamphile [1] and Luiz Fernando Cotica [2,*]**

[1] Environmental Biotechnology Program, State University of Maringá, Maringá 87020-900, Paraná, Brazil
[2] Department of Physics, State University of Maringá, Av. Colombo 5790, Maringá 87020-900, Paraná, Brazil
[3] Ingénierie des Matériaux Polymères, Université de Lyon, Université Claude Bernard Lyon 1, 69008 Lyon, France
[*] Correspondence: lfcotica@dfi.uem.br; Tel.: +55-3011-5904

**Abstract:** The use of a variety of microorganisms for the degradation of chemicals is a green solution to the problem of environmental pollution. In this work, fungi–magnetic nanoparticles were studied as systems with the potential to be applied in environmental remediation and pest control in agriculture. High food demand puts significant pressure on increasing the use of herbicides, insecticides, fungicides, pesticides, and fertilizers. The global problem of water pollution also demands new remediation solutions. As a sustainable alternative to commercial chemical products, nanobiocomposites were obtained from the interaction between the fungus *M. anisopliae* and two different types of magnetic nanoparticles. Fourier transform infrared spectroscopy, optical and electron microscopy, and energy dispersive spectroscopy were used to study the interaction between the fungus and nanoparticles, and the morphology of individual components and the final nanobiocomposites. Analyses show that the nanobiocomposites kept the same morphology as that of the fungus in natura. Magnetic measurements attest the magnetic properties of the nanobiocomposites. In summary, these nanobiocomposites possess both fungal and nanoparticle properties, i.e., nanobiocomposites were obtained with magnetic properties that provide a low-cost approach benefiting the environment (nanobiocomposites are retrievable) with more efficiency than that of the application of the fungus in natura.

**Keywords:** nanobiocomposites; biotechnology; environmental pollution removal

## 1. Introduction

The global population will reach 6 to 9 billion people by 2050 if current growth rates are maintained [1]. Thus, high food demands continue to put significant pressure on agriculture. Consequently, the use of herbicides, insecticides, fungicides, pesticides, and fertilizers (phytosanitary agents) would undoubtedly increase [2]. Phytosanitary agents, for example, induce undesired effects on soil and water, incorporating them in the food chain [3]. Further, there will be an increase in weeds and insects attaining immunity to these substances [4], causing billions of USD in liabilities due to crop losses [5–7]. Climatic changes, urbanization, and the unsustainable use of natural resources are other issues that the food-producing industry must handle. Another outcome is the acute global problem of water pollution, mainly in developing countries, due to the high use of phytosanitary agents. As a consequence, demand for high-quality water (i.e., water free of toxic chemicals and pathogens) has increased, leading to more powerful health-based regulations [8,9].

In this sense, the biological control of agricultural pests is in most cases preferable to chemical control [10]. Bioremediation using a variety of microorganisms for the degradation of chemicals is a green solution to the problem of environmental pollution. Mi-

croorganisms have been gifted by nature with the ability to degrade a wide spectrum of environmental pollutants.

Since there is an increase in the microbial monitoring of insect pests by entomopathogenic fungi in place of chemical control, there been a decrease in environmental impact in some plantations [11]. In particular, *Metarhizium anisopliae* is an entomopathogenic fungus that causes disease in insects and is globally employed as a biopesticide [12]. Due to the production of its infected units on a commercial scale, the ease of application in the field, low costs, and mainly its reduced environmental impact, the biological control of pests via *Metarhizium anisopliae* becomes relevant [13].

Another approach to deal with environmental treatments is the use of nanotechnological solutions. This research area has developed into a multidisciplinary field and has caused a revolution in the basic sciences (applied physics, chemistry, mechanics, biological and electrical engineering, robotics, and medicine) [13–15].

Regarding multiple suitabilities in nanotechnology, nanoparticles (particles at a nanometric scale) are being applied in many research areas and have contributed to finding solutions in the treatment of diseases, solar and wind energy, water treatment, mobile telephones, and numerous other products [16]. Focusing on environmental treatments, iron oxide nanoparticles have been applied for many different purposes.

In particular, due to their high chemical stability, the nanoparticles of magnetite ($Fe_3O_4$) are employed in several biological applications [17]. As an additional characteristic, magnetite nanoparticles hold magnetic properties [18]. Some biotechnological applications in use are the removal of pollutants [19], water staining [20], optimization in the use of fertilizers [21], and the capture of insects (*Aedes aegypti*, for example) when submitted to a magnetic field [22], which receive magnetic nanoparticles in the metathorax [22].

In this work, we used the properties of both microorganisms, especially fungi, and magnetic nanoparticles to create a multifunctional system with the potential to be applied in environmental remediation and pest control in agriculture.

The combination of nanoparticles and organisms or cells is called a biotemplate [23] or nanobiocomposite [19]. In other words, it is the combination of traditional strategies used for biological control and the capacity of nanomaterials in the control of insect pests. Consequently, studies are increasing on fungi to obtain nanobiocomposites with different applications, such as the removal of pollutants from water [19] and the biogenic synthesis of nanoparticles [24]. Li et al. [19] produced *Aspergillus* fungus and $Fe_3O_4$ nanoparticles. The cell wall of the *Aspergillus* fungus is formed by chitin and glucans, carbohydrates that have a large number of carboxyl groups in their composition. On the other hand, nanoparticles contain hydroxyls on their surfaces; thus, a dehydration reaction occurs, releasing water and allowing for the union between nanoparticles and the fungal wall. With this nanobiocomposite, the authors effectively removed radioactive uranium from water. In the end, the nanocomposite easily removed the material from the environment by using subsequent magnetic separation, a simple procedure that resulted in excellent collection [25]. Other works handling microorganisms adhering to nanomaterials were used for the bioremediation of contaminated environments [26–28].

Owing to the need for alternative research to decrease the amount of toxic agents in food production, and the effects due the interaction between nanoparticles and entomopathogenic fungi, this paper analyzes the biological interaction between magnetite nanoparticles ($Fe_3O_4$) and the entomopathogenic fungus *Metarhizium anisopliae* to obtain nanobiocomposites. In other words, the generation of nanobiocomposites of *Metarhizium anisopliae* and magnetite nanoparticles was studied, especially their microscopic morphology, how the connection between the fungus and the nanoparticles, and the retention of magnetization in the final nanobiocomposites occur.

## 2. Materials and Methods

### 2.1. Biological Material

Entomopathogenic fungus *Metarhizium anisopliae* (Metschnikoff) Sorokin was taken from the collection of microorganisms of the Microbial Biotechnological Laboratory (BIOMIC) at the State University of Maringá (UEM), Maringá-PR Brazil.

### 2.2. Magnetite Nanoparticles

$Fe_3O_4$ nanoparticle synthesis was fully described in previous works [29,30]. Two different methods were adopted: the polyvinyl alcohol (MNP-PVA) and coprecipitation (MNP-COP) synthesis routes. MNP-PVA nanoparticles were synthesized by adding iron (III) nitrate in a solution of PVA 10%. The mixture was warmed to start the dehydration and decomposition of PVA. A pyrolytic reaction at 500 °C produced magnetite nanoparticles. The coprecipitation synthesis was carried out via a combination of iron (II) and iron (III) salts in a water solution, followed by precipitation through the addition of ammonium hydroxide at 80 °C.

### 2.3. Nanobiocomposites (NBCs)

Nanobiocomposites (NBCs) were obtained from entomopathogenic fungus *M. anisopliae* and two different types of previously retrieved magnetic nanoparticles, MNP-COP and MNP-PVA. Conidia suspension was prepared in a water solution with Tween 80® (0.01% $v\,v^{-1}$), at a concentration of $2.690 \times 10^7$ conidia/mL. Next, 300 μL from this suspension was inoculated in glycerol and peptone-modified minimal medium under 110 RPM stirring during 3 days at 28 °C. Subsequently, the modified medium was centrifuged at 1500 RPM for 15 min. Lastly, the mycelium was filtered and settled on a Petri plate.

In the second step, portions of 0.25 g (wet weight) of *M. anisopliae* mycelium were incubated in different Erlenmeyer flasks filled with 50 mL of modified minimal medium. Different treatments inoculated these separated portions in a randomized scheme: (1) culture medium with 0.05 g of MNP-PVA, and (2) culture medium with 0.05 g of MNP-COP. As the control, some distinct suspensions were considered: minimal culture medium, culture medium with 0.05 g of MNP-PVA, and culture medium with 0.05 g of MNP-COP.

After keeping the treatments shaken for 2 days, and the Erlenmeyer flasks at 110 RPM and 28 °C, mycelium and bionanocomposites were washed for five times in deionized water, centrifuged at 1500 RPM for 15 min, and filtered. The final material was left to rest at 4 °C for 24 h. Afterwards, samples were prepared for morphological and magnetization investigations.

### 2.4. Fourier Transform Infrared Spectroscopy (FTIR)

Infrared spectra were recorded with a Thermo Scientific Nicolet iSO10 FTIR spectrophotometer. The powdered samples were milled with KBr and pressed into pellets. FTIR spectra in the range of 4000–400 cm$^{-1}$ were recorded with an accumulation of 200 scans and a resolution of 2 cm$^{-1}$.

### 2.5. Optical Microscopy

Fungi were submitted to a microcultural methodology that consisted of placing a block of agar on a sterile slide inside a sterile Petri plate. The fungal spores were seeded in the corners of the blocks and subsequently covered with sterilized coverslips. After 7 days in an oven at 28 ° C, the coverslips were carefully lifted, and fungi were examined. The images were analyzed under an Omicron Medical microscope and captured in an Axiocam Mrc.

### 2.6. Scanning Electron Microscopy and Energy-Dispersive X-ray Spectroscopy

*M. anisopliae* and NBCs were morphologically characterized via scanning electron microscopy (SEM). To prepare the dried samples for SEM analyses, the fungus (control) and NBCs were added in glass slides by using glutaraldehyde. After that, dehydration

was performed in a sequence of different ethanol concentrations (30, 50, 70, 80, 90, 95, and 100%). Fragments were dried in a $CO_2$ critical-point dryer (BAL-TEC CPD 030). Lastly, the obtained material was placed in aluminum stubs using conducting adhesive bands for electron microscopy and gold-sputtered to obtain a top conductive layer. A secondary electron detector was used to obtain the SEM images in an FEI Quanta 250 electron microscope.

To support the existence of magnetite nanoparticles on fungal surfaces, energy-dispersive X-ray spectroscopy (EDS) was employed to identify different chemical elements in the sample. Spectroscopic analyses were performed with an Oxford Instruments NanoAnalysis detector inside the electron microscope.

### 2.7. Transmission Electron Microscopy

The morphology and particle size analyses of magnetite nanoparticles were conducted using a JEOL JEM-1400 transmission electron microscope.

### 2.8. Nanobiocomposite Magnetization Test

Magnetic hysteresis loops were performed at room temperature in a homemade vibrating sample magnetometer by sweeping the applied magnetic field up to 15 kOe.

### 3. Results and Discussion

Magnetic nanoparticles from two distinct routes were used to obtain the nanobiocomposites. Typical TEM images for magnetite nanoparticles obtained with the coprecipitation (MNP-COP) and polyvinyl alcohol (MNP-PVA) synthesis routes are shown in Figure 1. Uniform near-spherical nanoparticles were produced. The mean particle size for MNP-COP was 28.4 nm; for MNP-PVA, it was 9.2 nm (see histograms in the inset of Figure 1).

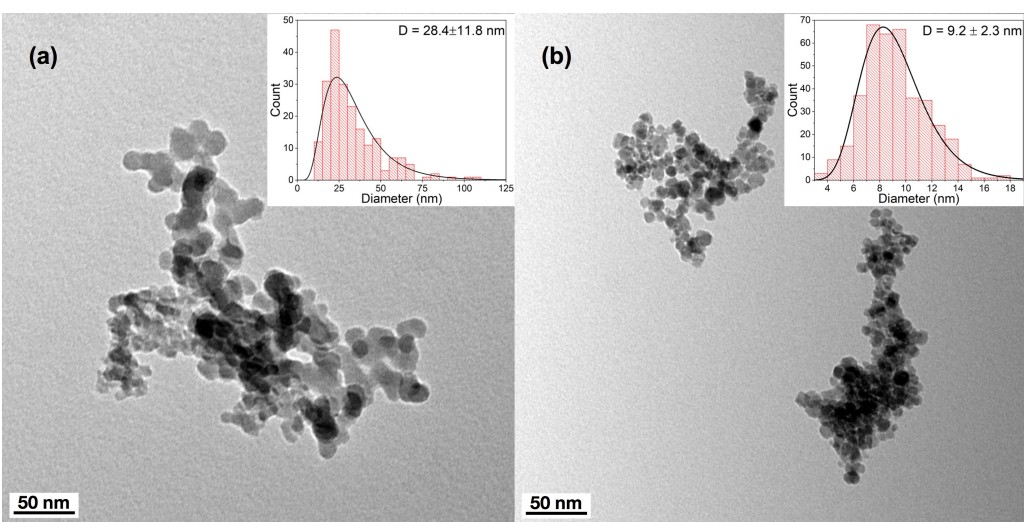

**Figure 1.** Transmission electron microscopy images for (**a**) MNP-COP and (**b**) MNP-PVA nanoparticles.

The morphology of *M. anisopliae* (Figure 2a), when observed in a light microscope (Figure 2b), revealed a filamentous organism with hyaline and septate mycelium featuring conidiophores from which cylindrical conidia emerge. Filamentous structures comprise tubular multicellular elements called hyphae (Figure 2b). *M. anisopliae* hyphae are septate, and the set of hyphae is called the mycelium. Reproduction produces asexual spores called conidia (Figure 2c).

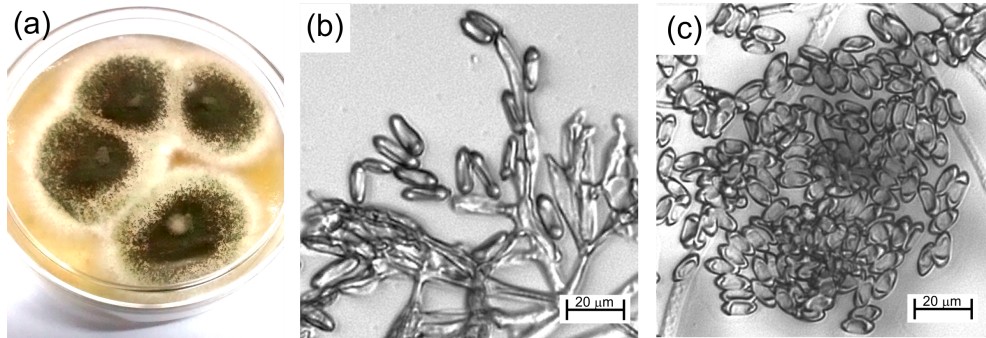

**Figure 2.** (**a**) *Metarhizium anisopliae*. (**b**,**c**) Fungal morphology observed in the light microscope.

The color and morphological differences between *M. anisopliae* and the nanobiocomposites (NBCs) obtained using MNP-COP and MNP-PVA are shown in Figure 3. Macroscopically, there were no significant changes in the morphology of NBCs when contrasted with the in natura fungus. The presence of magnetite nanoparticles turns NBCs dark gray, which is a typical color for magnetite nanoparticles, i.e., it is indicative of the presence or adherence of nanoparticles.

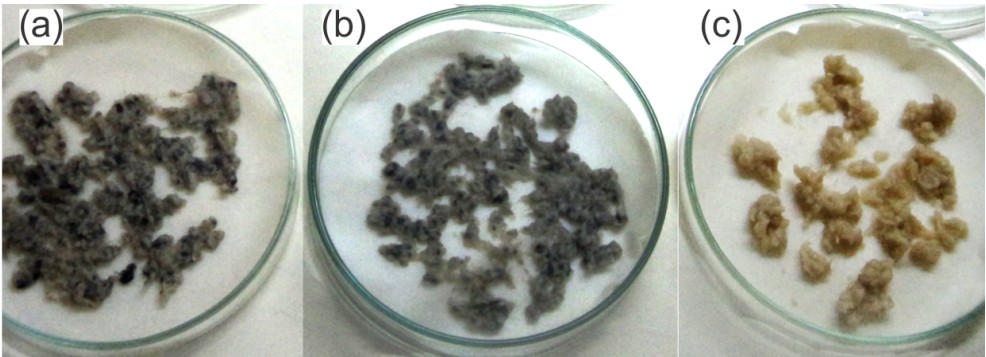

**Figure 3.** Nanobiocomposites obtained using (**a**) MNP-COP and (**b**) MNP-PVA nanoparticles, and (**c**) in natura *M. anisopliae.*

The FTIR spectra of $Fe_3O_4$ nanoparticles (MNP-COP and MNP-PVA) and NBCs (MNP-COP-NBC and MNP-PVA-NBC) are shown in Figure 4. For both types of iron oxide nanoparticles (Figure 4a), the absorption band at 441 cm$^{-1}$ was assigned to Fe–O stretching in octahedral sites, and the absorption bands at 582 and 630 cm$^{-1}$ were assigned to Fe–O stretching on octahedral and tetrahedral sites in the crystalline lattice of $Fe_3O_4$ [29,31,32]. The presence or adherence of nanoparticles is also indicated.

Figure 4b shows the FTIR spectrum of the in natura *M. anisopliae* sample with a typical fungal structure [33]. In the higher-wavenumber region, the characteristic band at ~3400 cm$^{-1}$ was assigned to the vibrations of the O–H of water and N–H. Absorption bands at 2925 and 2852 cm$^{-1}$ refer to the asymmetrical and symmetrical stretching of C–H in $CH_2$ mainly due to the lipids' absorbance [34,35]. The absorption band at 1747 cm$^{-1}$ due to C=O stretching was also from groups in lipids. The absorption bands at 1650 and 1552 cm$^{-1}$ were assigned to the C=O stretching vibrations of amide I, and the N–H bending and C–N stretching vibrations of amide II, respectively, and indicate the presence of proteins in the fungal structure. The characteristic band at 1385 cm$^{-1}$ was assigned to the symmetrical stretching of $COO^-$ group. The regions with a wavenumber in the range of 1200–900 cm$^{-1}$ were dominated by absorption bands of carbohydrates present in the cell wall [33].

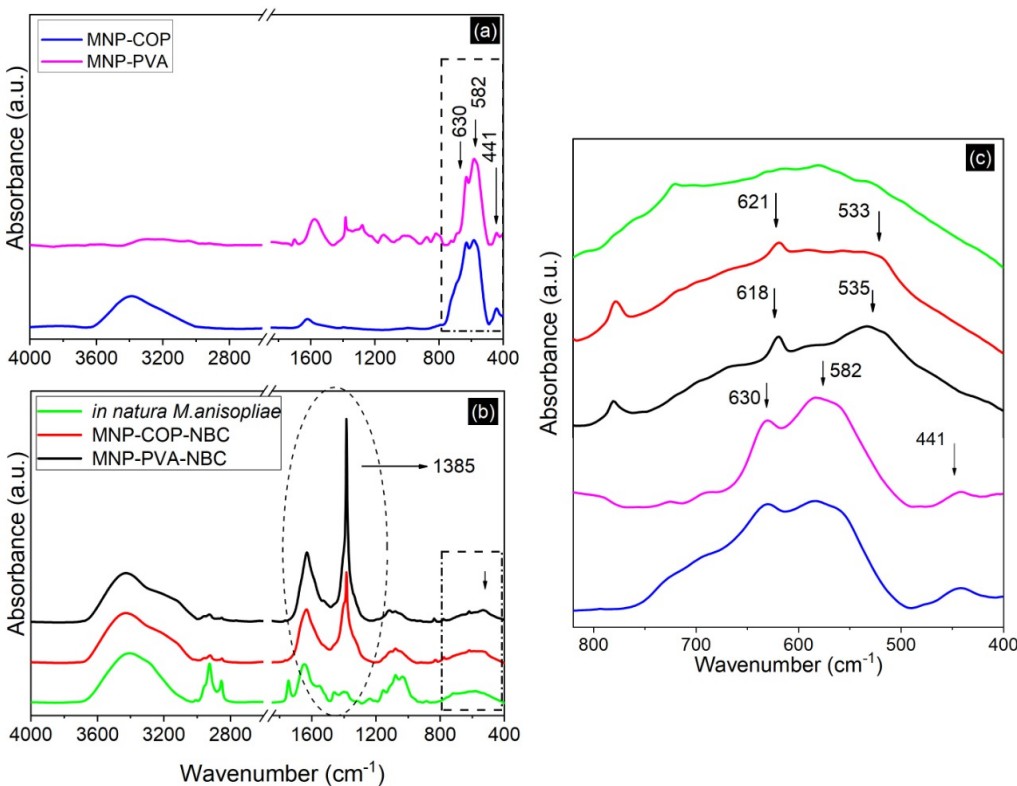

**Figure 4.** Fourier transform infrared (FTIR) spectra: (**a**) MNP-COP and MNP-PVA; (**b**) MNP-PVA NBC, MNP-COP NBC and in natura *M. anisopliae*; (**c**) details of (**a**) and (**b**) in the range of 820 cm$^{-1}$ to 400 cm$^{-1}$ for all samples.

For both NBCs in Figure 4b, it is possible to observe an intensity increase in the absorption band of the symmetrical stretching of $COO^-$ placed at 1385 cm$^{-1}$ in relation to the in natura sample. This indicates the formation of Fe–carboxylate complexes on the iron oxide surface of the nanoparticles [25,33].

Figure 4c shows details of the regions with a wavenumber below 800 cm$^{-1}$ for all samples. Absorption bands appeared at 621 and 533 cm$^{-1}$, and at 618 $^{-1}$ and 535 cm$^{-1}$ for the MNP-COP-NBC and MNP-PVA-NBC samples, respectively. This indicates that the Fe–O binding bands were present in the samples and overlapped with the fungal broad band in this region.

Different works demonstrated that filamentous fungi may undertake new chemical bonds with metals because the cell walls of these microorganisms have great bonding and intercellular absorption capacity with metals [19,36].

In their cell walls, fungi have a great number of complex organic compounds and their polymers, such as glucan (28%), polysaccharides (31%), proteins (13%), lipids (8%), and chitin and chitosan (2%). Studies on fungal biomass and algae suggested a dominant role of a metal bond ionized to carboxyl, phosphate, and amino groups. In fact, these functional groups are the sites of metal adsorption. Consequently, fungi survive metal toxicity through produced mechanisms as a direct response to the analyzed metals. Several authors reported that the *A. aculeatus* fungus has a nickel tolerance of up to 1473 mg/L [37,38].

Previous studies reported that the hydroxyl and carboxyl groups of the fungus *Penicillium sp.* interact with Fe–O bonds of magnetite, a compound and stable structure capable of remaining intact in a hostile environment. In fungus–$Fe_3O_4$ nanocomposites, nanoparticles are distributed uniformly on the surface of *Penicillium sp.* without any aggregation. These results are similar to those achieved in this work, i.e., fungus (entomopathogenic *M. anisopliae*)–$Fe_3O_4$ nanoparticle nanocomposites were obtained.

To clarify the influence of $Fe_3O_4$ nanoparticles on fungal morphology and nanobiocomposite formation, studies using scanning electron microscopy (SEM) were performed.

*M. anisopliae* was analyzed with SEM, and the images helped in observing the microstructural morphology (Figure 5a). Some dehydrated hyphae of *M. anisopliae* with typical fungal morphology were observed due to the preparation process for observation in electron microscopy.

Simultaneously, in situ energy-dispersive X-ray spectroscopy (EDS) was used to map the chemical elements in the material (Figure 5b). This methodology exposed the lack of any trace of iron (e.g., $Fe_3O_4$ nanoparticles) and reinforced the presence of other elements, such as carbon (blue points), oxygen (green points), and gold (from the coating process).

A SEM image of the nanobiocomposite obtained employing MNP-COP can be seen in Figure 6a. Some hyphae with regular morphology were observed, even after nanoparticles had adhered to the fungal structures, confirming that an NBC was obtained. As seen in Figure 6c, visible peaks related to iron cannot be reliably detected. This is because EDS analysis is performed on a narrow area or volume of a sample. Due to this key reason, we mapped some elements in the overall image (Figure 6b) to unequivocally verify the presence of iron in the sample and thereby that of magnetic nanoparticles. Thus, analysis with the total EDS scanning of NBC samples (Figure 6b) revealed that magnetic nanoparticles adhered to *M. anisopliae* hyphae. Green and blue points in the map of the MNP-COP NBC are carbon and iron atoms, respectively, evidencing that magnetite nanoparticles adhered to several regions of the sample. Gold from the coating process was also observed.

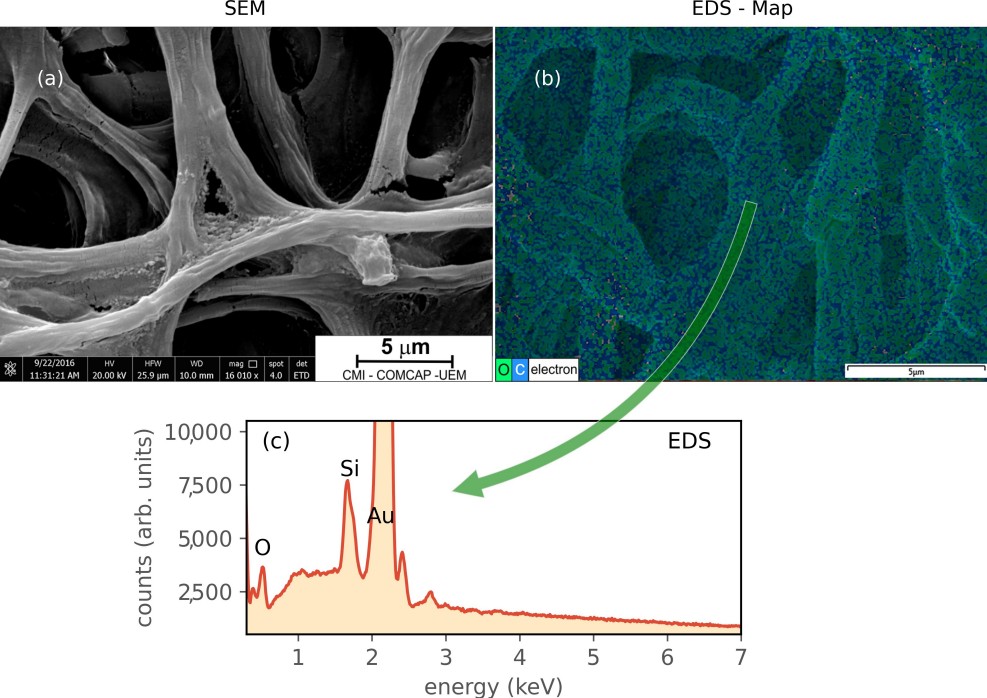

**Figure 5.** (**a**) Scanning electron microscopy (SEM) image and (**b**) energy-dispersive X-ray spectroscopy (EDS) analysis of *M. anisopliae*. (**c**) EDS analysis at the selected point.

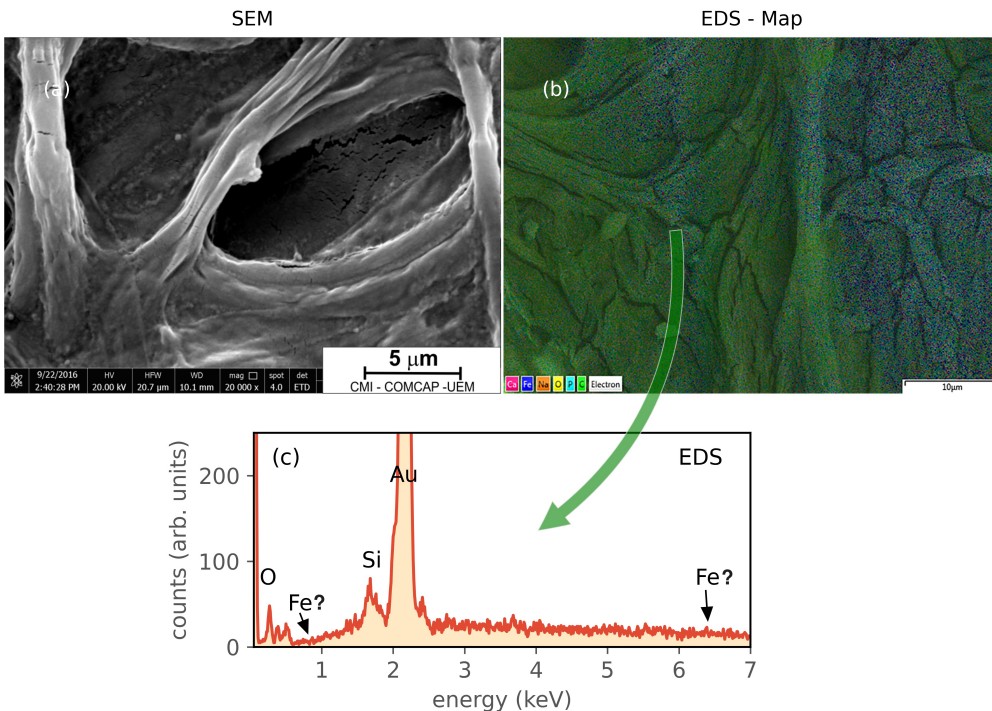

**Figure 6.** (**a**) Scanning electron microscopy (SEM) image and (**b**) energy-dispersive X-ray spectroscopy (EDS) analysis of MNP-COP nanobiocomposite. (**c**) EDS analysis at the selected point.

Almost the same observations could be obtained from the SEM and EDS analyses of MNP-PVA NBC (Figure 7a,b). Figure 7 shows that we were able to verify the effective presence of iron in both the EDS spectrum (Figure 7c) and the image mapping (Figure 7b). In Figure 7b, red and blue points denote iron and carbon atoms, respectively, and confirm the presence of adhered $Fe_3O_4$ nanoparticles. Again, gold from the coating process was observed.

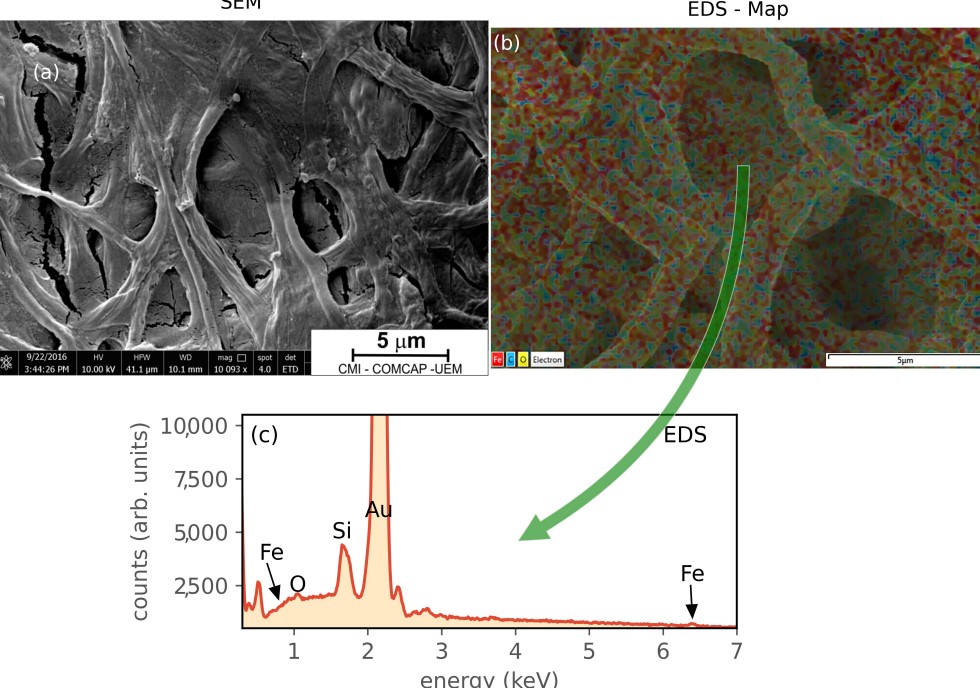

**Figure 7.** (**a**) Scanning electron microscopy (SEM) image and (**b**) energy-dispersive X-ray spectroscopy (EDS) analysis of MNP-PVA nanobiocomposite. (**c**) EDS analysis at the selected point.

As a conclusion, following the spectroscopic and microstructural results, *M. anisopliae*, MNP-COP NBC, and MNP-PVA NBC exhibited similar vegetative growth with the same nature of hyphae and colony, and reproductive growth with conidia arrangement in a conidiophore, which is characteristic of the species. In this sense, nanoparticles did not interfere with fungal growth and morphology (Figure 8).

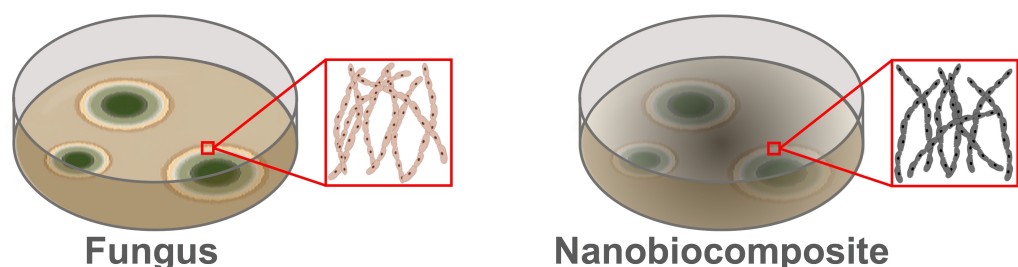

**Figure 8.** Magnetite nanoparticles did not interfere with fungal growth and morphology.

In fact, the surface properties of nanoparticles are important in interactions with biological systems such as fungi [39]. In the current study, nanobiocomposite samples with MNP-COP and MNP-PVA, and evaluated with SEM and EDS had magnetic nanoparticles adhered to the hyphae of the mycelium grown in the culture medium [40].

Research on gold nanoparticles and *Aspergillus sp.* to obtain microfilaments revealed that gold nanoparticles organize themselves on the hyphae surface according to fungal growth [41]. Schwegmann et al. found that the amount of iron oxide nanoparticles covering the surface of microorganisms was related to electrostatic interaction forces [42]. The same conclusions can be used to explain the *M. anisopliae*-$Fe_3O_4$ nanoparticles interaction in this work.

A further advantage of employing $Fe_3O_4$ nanoparticles to form the nanobiocomposite is that NBC may be easily removed from an aquatic environment with magnetic separation [19]. This fact was also examined in the current study.

A way to show NBC magnetic properties is subjecting them to a magnetic field. Magnetization curves (at room temperature) for MNP-COP, MNP-PVA, and NBCs are shown in Figure 9. All samples exhibited super-paramagnetic-like (SPM) behavior with almost zero coercivity.

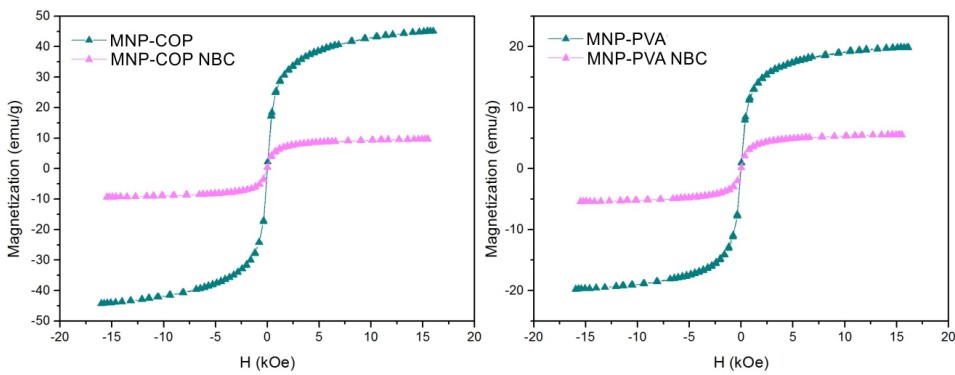

**Figure 9.** Magnetization curves (at room temperature) for MNP-COP, MNP-PVA, and NBCs.

The obtained saturation magnetization (Ms) values were 45.0 and 19.8 emu/g for MNP-COP and MNP-PVA, respectively. These values were lower than those reported for multidomain bulk magnetite (∼90 emu/g) [29,43]. In addition to the nanoparticles being produced by different synthesis methods, a decrease in Ms is often attributed to a reduction in particle size, and correlated to surface spin contributions and surface defects [44]. On the other hand, TEM images (Figure 1) show that the mean particle size of MNP-COP was greater than that of MNP-PVA. This can explain the higher Ms value for MNP-COP, and

leading MNP-COP NBC to having a better response to an applied magnetic field when compared to that of MNP-PVA NBC.

In the case of NBC samples, the magnetization curves were very similar to those of the nanoparticles, with the reduction in magnetization due to the presence of the fungus, which is an organic compound, involving the magnetic nuclei as a shell.

A way to assess the NBC response to a magnetic field is bringing them closer to a magnet. In this work, the nanobiocomposites were brought close to Nd–Fe–B magnets. Figure 10 shows that the NBCs were attracted by the magnets, confirming that the NBCs had magnetic properties and could be removed from a liquid medium by using a magnetic field. As MNP-COP NBC showed higher $M_S$, it had a better efficiency in the separation from aqueous environments.

In summary, the employment of nanobiocomposites with magnetic nanoparticles is a low-cost multifunctional approach benefiting the environment (nanobiocomposites are retrievable) with better efficiency than the application of fungi in natura. The number of nanoparticles that adhered to fungal hyphae could not be precisely identified using the techniques used in this work. The main objectives of this work were to study the interaction between nanoparticles and hyphae, and the magnetic control of nanobiocomposites.

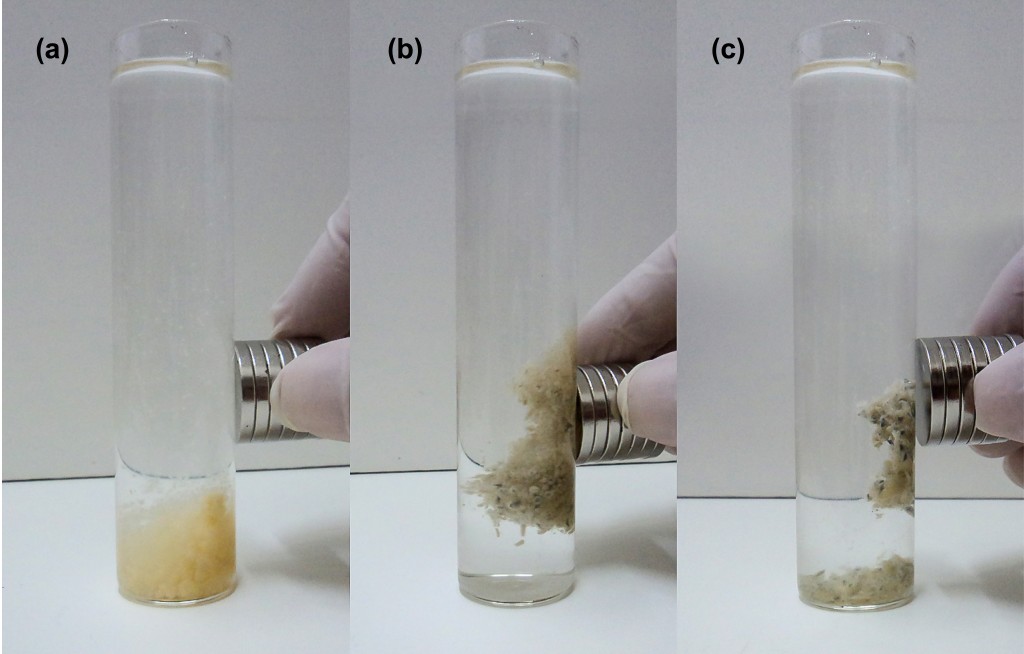

**Figure 10.** Nanobiocomposites were exposed to magnetic fields produced by Nd–Fe–B magnets: (**a**) in natura *M. anisopliae*, (**b**) MNP-COP nanobiocomposite, and (**c**) MNP-PVA nanobiocomposite.

## 4. Conclusions

In this work, *M. anisopliae*/Fe$_3$O$_4$ magnetic nanoparticles' direct interaction as nanobiocomposites was studied as an arrangement with potential application in environmental remediation and pest control in agriculture. Spectroscopic and microscopic studies were carried out to carefully explore the interaction between fungus and nanoparticles, and the comparative morphology of individual components and the final nanobiocomposites. FTIR spectroscopic results precisely indicate the presence or adherence of the nanoparticles. Hyphae with regular morphology were observed even after nanoparticles had adhered to the fungal structures. The energy-dispersive X-ray spectroscopic analysis of NBCs revealed that magnetic nanoparticles had adhered to fungal hyphae. *M. anisopliae*, MNP-COP, and MNP-PVA nanobiocomposites exhibited similar vegetative growth with the same hyphae and colony. In this sense, nanoparticles did not interfere with fungal growth and morphology. All these consistent results confirm that nanobiocomposites were properly obtained.

Magnetic measurements attested to the magnetic properties of the nanobiocomposites. MNP-COP had higher magnetization than that of MNP-PVA. In this sense, MNP-COP NBC can be more efficient in removing aqueous environments. The analyses showed that the intended interaction between the *M. anisopliae* fungus and $Fe_3O_4$ was successfully achieved, keeping the same morphology as the fungus in natura as shown with the electron microscopy analyses. In conclusion, nanobiocomposites were studied as systems with considerable potential to be applied in environmental remediation and pest control in agriculture as a sustainable alternative to commercial chemical products. These nanobiocomposites possess both fungal and nanoparticle properties. We obtained nanobiocomposites with magnetic properties provide a low-cost approach, benefiting the environment (are retrievable) with better efficiency than that of the application of the fungus in natura. Both nanobiocomposites had properties of environmental remediation and pest control in agriculture, but the MNP-COP nanobiocomposite presented better magnetic properties. Further uses for magnetic nanobiocomposites are already under study, mainly in the bioremediation process, such as in water contaminated with dyes and pesticides. We hope to publish new results in the near future. Lastly, the low cost is evident in three key aspects: the low cost of the nanobiocomposite synthesis, the comparative ease of properly obtaining the fungus, and the reduced cost of the nanoparticles used in this work.

**Author Contributions:** T.d.O.C., NBC assembly, biological experiments, SEM/EDS experiments, and text writing. R.D.B., nanoparticle synthesis, FT-IR measurements, and text contributions. V.A.d.O.J., NBC assembly, and biological and SEM/EDS experiments. A.D.P., NBC assembly and biological experiments. A.G., NBC assembly and biological experiments. P.N.d.O., nanoparticle synthesis and text contributions. G.S.D., magnetic measurements and text contributions. I.A.d.S., magnetic measurements and text contributions. L.F.C. project administration, formal analysis, funding acquisition and text writing. J.A.P., conceptualization, methodology, and supervision. All authors have read and agreed to the published version of the manuscript.

**Funding:** This research was funded by Coordenação de Aperfeiçoamento de Pessoal de Nível Superior: Fellowship funding; Conselho Nacional de Desenvolvimento Científico e Tecnológico: Project Funding and fellowship funding; Fundação Araucária de Apoio ao Desenvolvimento Científico e Tecnológico do Estado do Paraná: Project Funding; Financiadora de Estudos e Projetos: Project Funding.

**Data Availability Statement:** Not applicable.

**Acknowledgments:** The authors would like to acknowledge the CNPq, Capes, Fundação Araucária de Apoio ao Desenvolvimento Científico e Tecnológico do Estado do Paraná and Finep Brazilian funding agencies.

**Conflicts of Interest:** The authors declare no conflict of interest.

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
