# Peer review of "Fungus-Based Magnetic Nanobiocomposites for Environmental Remediation"

_magnetochemistry, doi:10.3390/magnetochemistry8110139_

Round 1

Reviewer 1 Report

The authors' studies are relevant and of considerable interest at the present time. The manuscript is noteworthy and may be printed.

However, it is necessary to eliminate a number of questions and comments in order for the results of the study to become clearer.

I indicated all the comments in the text of the manuscript.

Author Response

Dear Reviewer,

We thank you for revising our submitted manuscript ‘‘Fungus based magnetic nanobiocomposites for environmental remediation’’ (magnetochemistry-1877123). The questions were carefully analyzed, and the manuscript was modified (improved). Please find below our specific answers.

The authors' studies are relevant and of considerable interest at the present time. The manuscript is noteworthy and may be printed.

However, it is necessary to eliminate a number of questions and comments in order for the results of the study to become clearer.

I indicated all the comments in the text of the manuscript.

1- It seems to me that examples of the use of nanobiocomposites should be described in more detail. As well as the problems that arise with this approach.

Thanks for the comment. We improved the text.

2 - How many replications were used in the experiments, was statistical processing carried out?

Triplicates were made, but we did not analyze this material statistically.

3- MPN-PVA?

Thanks for the comment. We made the change to the text.

4- M. anisopliae

Thanks for the comment. We made the change to the text.

5- In figure 4a and 4b, it is not entirely clear what these bands are? You can indicate in the text which bands, characteristic of nanoparticles, stand out in the NBC spectra.

We thank the reviewer for the suggestion. We improved the discussion and the FTIR figure.

6- It would be better to use real photographs of colonies under a microscope, because this figure does not support the conclusion.

We appreciate what the reviewer suggested. However, we believe the figure presented is more illustrative. So we are going to maintain this figure.

7-It is necessary to correct the caption to the figure and the designations in the figure: NBC-COP on MNP-PVA

Thanks for the comment. We made the change in the caption.

8- This does not apply to Conclusions, because already been in the Introduction.

Thanks for the comment. We made changes to improve the text.

9- The article does not confirm this.

The direct interaction between the fungus and the nanoparticles can be confirmed by FTIR measurements. And the morphological studies are clearly based on electron microscopy images. There are no apparent changes in fungal morphology after successful interaction with the nanoparticles.

10- Do the resulting nanobiocomposites have any other advantages, apart from being extractable from the aquatic environment? 

We are still exploring other key aspects. As a practical example, we are investigating the observed increase in the effectiveness of these fungi in an innovative treatment of textile dyes.

How do nanopurges affect the effectiveness of microscopic fungi? 

Unfortunately we don't have this answer right now. We are still studying this aspect.

Where and how can such nanobiocomposites be used?

These nanobiocomposites can be used mainly in the bioremediation process. For example in water contaminated with dyes and pesticides.

How was the low-cost of this approach assessed?

The low cost is present in three key aspects: the low cost of the nanobiocomposite synthesis, the comparative ease of properly obtaining the fungus, and the reduced cost of the nanoparticles used in this work.

Reviewer 2 Report

The paper deals about the preparation of new bionanocomposites based on fungus M. anisopliae and Fe3O4 nanoparticles that were obtained from two different synthetic methods. The goal of the manuscript is clear and the interest justified. The usual methodology has been applied to perform a basic characterization of the new systems. Different points must be taken into account before acceptance:

1.       The percentage of magnetic nanoparticles incorporated into fungi should be explicitly indicated (try also to improve lines 103-104).

2.       I suggest splitting the results and discussion section in several subsections.

3.       Discussion of FTIR spectra should be improved. Bands associated to magnetic particles should be signalled in the spectra of the bionanocomposites. Peaks at 535 539 and 441 cm-1 should be discussed in the text.

4.       Discussion concerning the presence of iron particles in the MNP-COP derived sample should be improved and the presence of these particles in the corresponding bionanocomposite better supported.

5.       Figure 9: The legend seems wrong in the plot of the right (MNP-PVA instead on MNP-COP?). Can the authors give some quantitative data from magnetization experiments about the differences between the two studied bionanocomposites.

6.       Conclusions should incorporate some quantitative data supporting the differences between the two evaluated bionanocomposites.

7.       Grammatical and typing mistakes should be corrected.

Author Response

Dear Reviewer,

We thank you for revising our submitted manuscript ‘‘Fungus based magnetic nanobiocomposites for environmental remediation’’ (magnetochemistry-1877123). The questions were carefully analyzed, and the manuscript was modified (improved). Please find below our specific answers.

The paper deals about the preparation of new bionanocomposites based on fungus M. anisopliae and Fe3O4 nanoparticles that were obtained from two different synthetic methods. The goal of the manuscript is clear and the interest justified. The usual methodology has been applied to perform a basic characterization of the new systems. Different points must be taken into account before acceptance:

  1.       The percentage of magnetic nanoparticles incorporated into fungi should be explicitly indicated (try also to improve lines 103-104).

Theoretically, 0.05 g of nanoparticles were added for every 0.25 g of wet fungus. However, we cannot say for sure how much of these particles adhered to the fungus. We need more studies to answer this question. However, this does not interfere with our conclusions in this paper. Here the main objective is to study the interaction of nanoparticles with the fungus and magnetically control the fungi removal from an aqueous medium.

  1.       I suggest splitting the results and discussion section in several subsections.

We thank the reviewer for the comment. But, in our opinion, the format presented in the manuscript can lead the reader to better understand the findings.

  1.       Discussion of FTIR spectra should be improved. Bands associated to magnetic particles should be signalled in the spectra of the bionanocomposites. Peaks at 535 539 and 441 cm-1 should be discussed in the text.

We thank the reviewer for the suggestion. We improved the discussion and the FTIR figure.

  1.       Discussion concerning the presence of iron particles in the MNP-COP derived sample should be improved and the presence of these particles in the corresponding bionanocomposite better supported.

We thank the reviewer for the suggestion. We improved the discussion by adding text to lines 156 and 161.

However, some good discussion is already in the manuscript. For example: “analysis by total EDS scanning of NBC sample (Figure 6b) reveals that magnetic nanoparticles have adhered to M. anisopliae hyphae. Blue points in the map of the MNP-COP NBC assign iron atoms, evidencing magnetite nanoparticles adhered in several regions of the sample.”

  1.       Figure 9: The legend seems wrong in the plot of the right (MNP-PVA instead on MNP-COP?). Can the authors give some quantitative data from magnetization experiments about the differences between the two studied bionanocomposites.

Thanks for the comment. We fixed the caption and made changes to insert comparisons in the text.

  1.       Conclusions should incorporate some quantitative data supporting the differences between the two evaluated bionanocomposites.

Thanks for the comment. We made changes to suit the findings.

  1.       Grammatical and typing mistakes should be corrected.

Thanks for the comment. We made changes to suit the paper.

Round 2

Reviewer 2 Report

The authors have only partially taken previous comments into account.

Author Response

Dear Reviewer,

We would like to thank you again for revising our submitted manuscript ‘‘Fungus based magnetic nanobiocomposites for environmental remediation’’ (magnetochemistry-1877123). Your question was carefully analyzed, and the manuscript was modified (improved). Please find below our specific answer.

- The percentage of magnetic nanoparticles incorporated into fungi:

We understand the reviewer questioning. However, the number of nanoparticles adhered to fungal hyphae cannot be precisely identified using the techniques used in this work. On the other hand, the main objectives of this work were to study the interaction between nanoparticles and hyphae and the magnetic control of nanobiocomposites. This text has been added to the manuscript text for clarity.

- Conclusions:

The text of the conclusions has been improved.